# Association between different types of mass media and antenatal care visits in India: a cross-sectional study from the National Family Health Survey (2015–2016)

Dhriti Dhawan ,[1] Ramya Pinnamaneni ,[2] Mesfin Bekalu ,[2] Kasisomayajula Viswanath  [1,2]

[1]Medical Oncology, Dana-Farber Cancer Institute, Boston, Massachusetts, USA
[2]Social and Behavioral Sciences, Harvard T.H Chan School of Public Health, Boston, Massachusetts, USA

**Correspondence to**
Dr Dhriti Dhawan;
dhriti_dhawan@dfci.harvard.edu

## ABSTRACT

**Objective** To generate evidence for the association between different types of mass media and antenatal care (ANC) visits in India.

**Design** A cross-sectional study design, analysing data from India's National Family Health Survey 4 (NFHS-4), 2015–2016.

**Setting** Rural and urban India.

**Participants** From NFHS-4, women who had given birth in the last 5 years before survey administration were included in this study. Women with missing information about their number of ANC visits and their caste were excluded, leaving 187 894 women in the final analytical sample.

**Primary outcome measures** Logistic regression analysis was conducted to determine the association of ANC utilisation with mass media exposure.

**Results** Overall, our study showed that high exposure to all four types of mass media was positively associated with making at least eight ANC visits. In rural India, women who had high exposure to newspaper/magazine (adjusted OR (aOR), 1.43; 95% CI, 1.31 to 1.57), radio (aOR, 1.22; 95% CI, 1.09 to 1.37), television (aOR, 2.07; 95% CI, 1.94 to 2.2) and movies (aOR, 1.33; 95% CI, 1.2 to 1.47) were more likely to make at least eight ANC visits. In urban India, women who had high exposure to newspaper/magazine (aOR, 1.12; 95% CI, 1.02 to 1.24), radio (aOR, 1.37; 95% CI, 1.13 to 1.65), television (aOR, 1.39; 95% CI, 1.24 to 1.55) and movies (aOR, 1.23; 95% CI, 1.09 to 1.38) were more likely to make at least eight ANC visits.

**Conclusions** Our findings emphasise the need for increased awareness about adequate ANC visits in India, to improve maternal, neonatal and child health outcomes. Our study highlights that television penetration is broader than other forms of media and has the potential to create awareness about health in both urban and rural populations. These findings can inform ANC-related health awareness campaigns in the country to allocate resources to appropriate media sources to encourage healthy behaviours.

### Strengths and limitations of this study

► Our study provides a novel insight into the association between mass media exposure and making at least eight antenatal care (ANC) visits in India, with the central focus on mass media exposure.

► Our results are derived from a nationally representative survey and are generalisable to women in India.

► The data enable us to determine trends stratified by urbanicity, states, socioeconomic status, caste and religion, which can further help to narrow down at-risk populations and design customised interventions.

► The cross-sectional design of the data limits the ability to derive any causal inferences on the relationship between the variables. Nonetheless, the strong associations between media use and ANC visits suggest the potential promise of media in maternal health promotion.

► The data set has no information about the type of ANC-related media content available to women in India.

## INTRODUCTION

Antenatal care (ANC) is widely acknowledged as a critical tool in reducing stillbirths and complications in pregnancy and promoting positive pregnancy experience.[1] Components of ANC are risk identification, prevention and management of pregnancy-related or concurrent diseases, and maternal health education and health promotion.[1] ANC is conducive for a healthy pregnancy and improves the chances of survival for the mother as well as the child. It has also been shown to support the growth and development of the child until the age of 5 years.[2] Women who use ANC services are more likely to use other essential services such as counselling for breastfeeding, nutrition, postpartum family planning and childhood immunisation.[3]

Before 2016, the WHO recommended at least four ANC visits during normal pregnancy and that the first visit be completed within the first 8–12 weeks of conception.[1] However, in 2016, the WHO changed its recommendation to a minimum of eight ANC visits.[1] This was because, compared with four visits, eight or more ANC visits have the potential to reduce perinatal deaths by up to 8 per 1000 births, as it provides more opportunities for detecting and treating potential complications.[4] Despite these recommendations, women of low-income and middle-income countries have low ANC coverage.[1 5] According to UNICEF, there are wide disparities in ANC coverage across countries, with the lowest levels in sub-Saharan Africa and South Asia.[6] There is an urgent need to create awareness about the benefits of using ANC services and support interventions aimed towards increasing their utilisation to improve maternal, neonatal and child health. We need strategies that can promote ANC service utilisation at scale.

## Mass media campaigns

Mass media are critical tools in promoting health through two key strategies: (1) their widespread penetration promotes broad reach to key audiences across boundaries, and (2) exposure to specific messages in the media is known to shape public knowledge, attitudes, beliefs and behaviours.[7] Routine use of media can influence both prohealthy and antihealthy behaviours.[8] Media campaigns have been extensively used over the past few decades to induce behavioural changes in populations, especially in the context of substance use, such as tobacco, alcohol and illicit drugs.

Creating awareness through mass media can encourage positive or discourage negative health-related behaviours across populations, through direct and indirect pathways.[8] These behavioural changes occur passively as a result of routine media use.[9] Mass media campaigns can invoke cognitive or emotional responses at an individual level by informing the population about the benefits and risks of a particular health-related behaviour. These campaigns frequently associate emotions with achieving change, strengthening the probability of alteration and increasing the likelihood of adopting new behaviours.[10] For example, as a part of India's Pradhan Mantri Surakshit Matritva Abhiyan (PMSMA) campaign, public service announcements (PSAs) were released. These PSAs have prompted pregnant women and their family members to seek ANC during pregnancy, informed them about the benefits of these visits and highlighted that these visits will be free.[11] The WHO commended India for its groundbreaking progress in reducing maternal mortality and credited the PMSMA campaign for its contribution towards this achievement.[12]

Many low-income countries face poor child survival before 5 years of age because of inadequate treatment of diarrhoea, non-vaccination for preventable diseases, inadequate ANC and inadequate breastfeeding.[13 14] Mass media campaigns have been used to spread awareness and educate populations about each of these causes and have resulted in mixed evidence for success.[9] Television and radio are the most popular media for creating awareness among large audiences. However, print media, such as magazines and newspapers, and outdoor media, such as billboards and posters, have also proven effective.[9] In India, 65.2% of households own a television and 8.1% of households own a radio. Moreover, 66% of women in rural India and 92% women in urban India have regular exposure to some kind of mass media.[15] Broadcasting important messages related to ANC through mass media may help in increasing awareness at the population level.[2]

## ANC in India

Our study examines the association between mass media exposure and ANC visits in India. Indian women carry a heavy burden of poor nutritional status and infections, which is associated with high child mortality and maternal mortality.[16] In 2017, India recorded 35 000 maternal deaths—the second highest in the world.[17] In 2019, India recorded 882 000 under-5 deaths and 549 000 neonatal deaths—the highest in the world.[18] ANC visits are of high importance because they can help prevent adverse effects on the health of women and children, and reduce the risk of under-5 and neonatal deaths.[14] To encourage women to access ANC in India, public healthcare facilities provide these services for free.[15] Moreover, women of low socioeconomic status (SES) are given cash incentives for giving birth in a government or an accredited private healthcare facility through schemes such as *Janani Suraksha Yojana* (JSY). This scheme also incentivises healthcare workers to facilitate ANC visits along with institutional births.[19] Studies have shown that JSY has significantly increased ANC visits in India.[20] Despite such incentives, only 51.6% of pregnant Indian women received four or more ANC visits, which is lower than the global average of 62% during 2010–2016.[2 6]

Many intervention programmes in India use mass media as one of the communication tools to create health-related awareness.[21 22] For example, the National Nutrition Mission of India (*POSHAN Abhiyaan*) plans to use various platforms including mass media to inspire a people's movement, *Jan Andolan*, by creating awareness about nutrition and ensuring the engagement of the population at different levels.[23] The objective of our study is to generate evidence for the association between different types of mass media and ANC visits in India. Specifically, within ANC, we tested the association of the self-reported number of ANC visits with the level of exposure to four types of mass media: newspapers/magazines, radio, television and movies.

## METHODS
### Data source

This analysis uses individual-level data from India's National Family Health Survey 4 (NFHS-4), 2015–2016.[15] The NFHS series are nationally representative

cross-sectional surveys that provide data on a range of demographic, socioeconomic, maternal and child health outcomes; reproductive health; and family planning. NFHS-4 gathered information from 601 509 households, 699 686 women and 112 122 men, with a response rate of 97%. Of the women, 190 898 women had given birth in the last 5 years. In all, 1854 women were missing information about their number of ANC visits during pregnancy, and 1150 women were missing information about their caste. These women were excluded from the study, thereby leaving 187 894 women in the analytical sample.

## Sample design

The NFHS-4 used a two-stage stratified sampling design. The 2011 census served as the sampling frame for the selection of primary sampling units (PSUs). The samples were stratified by rural and urban areas, and villages were PSUs in the rural areas and census enumeration blocks were PSUs in the urban areas. Before the main survey, for each PSU, a household mapping and listing operation was carried out. Selected PSUs with at least 300 households were further segmented into 100–150 household segments. Systematic sampling was used to randomly select two of the segments, with probability proportional to segment size. Hence, an NFHS-4 cluster consists of either a PSU or a segment of a PSU. Twenty-two households were randomly selected with systematic sampling in every selected rural and urban cluster.

## Outcome measure

For their most recent pregnancy in the last 5 years, women were asked, 'How many times did you receive antenatal care during this pregnancy?' This was a continuous variable, which was recoded to a binary variable as women who had eight or more ANC visits and women who did not.

## Exposures

Women were asked 'Do you read a newspaper or magazine almost every day, at least once a week, less than once a week or not at all?' 'Do you listen to the radio almost every day, at least once a week, less than once a week or not at all?' and 'Do you watch television almost every day, at least once a week, less than once a week or not at all?' These were categorical variables with the four given option categories. These variables were recoded to dichotomous variables with response options 'high exposure' (for women who responded 'almost every day') and 'low exposure' (combining women who responded 'at least once a week', 'less than once a week' or 'not at all'). They were also asked, 'Do you usually go to a cinema hall or theatre to see a movie at least once a month?', with response options 'yes' and 'no'.

## Covariates

The age of the participants was categorised in increments of 5 years and ranged from 15 to 49 years. Educational attainment of the participants was categorised per the Indian educational system: no education, incomplete primary education, complete primary education, incomplete secondary education, complete secondary education and higher. Participants were grouped according to their marital status: never married, married, widowed, divorced and separated. In India, wealth has been established as a valid measure of SES. It can be defined as a measure of housing characteristics and possession of materials. Using principal component analysis, a wealth score was assigned to each participating household. Each person in the household was assigned a household score and ranked according to it. The distribution was divided into quintiles. The wealth of the participants was categorised accordingly.[15] The caste system in India has contributed to the social marginalisation of people who belong to certain groups and may potentially influence access to healthcare, including maternal healthcare.[24] NFHS reports four categories: scheduled castes, scheduled tribes, other backward classes and general. The urbanicity of the participants was categorised as rural or urban. The religion of the participants was categorised as Hindu, Muslim, Christian, Sikh and other.

## Statistical analysis

Frequencies of key outcome and exposure variables as well as the covariates were obtained to describe the study sample. Participants with 'do not know' responses for our variables of interest were recoded to missing and excluded from this study. Logistic regression analysis was conducted to determine the association between our outcome variable, 'at least eight ANC visits', and our respective exposure variables: 'newspaper/magazine', 'radio', 'television' or 'movie'. Multivariable logistic regression was conducted with each of the media variables while adjusting for other media variables and the covariates. The data were analysed and weighted using R V.3.6.1 (5 July 2019). Survey weights and clustering within PSUs were included in the analysis, and survey-weighted logistic regression was carried with the R survey package.

## Patient and public involvement

Patients and/or the public were not involved in the design, or conduct, or reporting, or dissemination plans of this research.

## RESULTS

Based on our analysis of the NFHS-4 survey, Indian women reported making an average of 4.8 ANC visits (SE ±0.03) during pregnancy. As per WHO's revised recommendation in 2016, only 20.3% of pregnant Indian women received at least eight ANC visits. In total, 9.9%, 3.3% and 55.1% of our target population had high exposure to newspaper/magazine, radio and television, respectively, and 7.4% of our target population went to cinema to watch a movie at least once a month (table 1). Needless to say, television penetration and reach are broad compared with other media. The percentage of women making at

| Table 1  Socioeconomic, demographic and media exposure characteristics by at least eight antenatal care visits among women who gave birth in the last 5 years, in the National Family Health Survey 4 (2015–2016) | At least eight antenatal care visits (N=29 812) |
|---|---|
| | (%*) |
| Total, N=187 894 (100.0%) | 20.3 |
| Newspaper/magazine | |
| Low exposure (90.0%) | 17.9 |
| High exposure (9.9%) | 40.8 |
| Radio | |
| Low exposure (96.6%) | 19.8 |
| High exposure (3.3%) | 33.2 |
| Television | |
| Low exposure (44.9%) | 9.5 |
| High exposure (55%) | 29.1 |
| Movie | |
| Less than once a month (92.6%) | 18.9 |
| At least once a month (7.4%) | 37.1 |
| Age (years) | |
| 15–19 (3.4%) | 18.8 |
| 20–24 (31.3%) | 19.8 |
| 25–29 (37.6%) | 21.8 |
| 30–34 (18.3%) | 20.9 |
| 35–39 (6.9%) | 16.7 |
| 40–44 (1.9%) | 11.4 |
| 45–49 (0.6%) | 6.2 |
| Educational attainment | |
| No education (27.6%) | 7.0 |
| Incomplete primary (5.9%) | 14.4 |
| Complete primary (7.4%) | 12.4 |
| Incomplete secondary (37.8%) | 23.8 |
| Complete secondary (9.2%) | 31.1 |
| Higher (12.0%) | 38.9 |
| Current marital status | |
| Never married (0.1%) | 16.8 |
| Married (98.6%) | 20.3 |
| Widowed (0.7%) | 17.5 |
| Divorced (0.1%) | 24.1 |
| Separated (0.4%) | 20.3 |
| Wealth | |
| First quintile (poorest) (23.4%) | 5.7 |
| Second quintile (21.2%) | 13.4 |
| Third quintile (19.9%) | 21.7 |
| Fourth quintile (19.0%) | 29.3 |
| Fifth quintile (richest) (16.6%) | 37.4 |

Continued

| Table 1  Continued | At least eight antenatal care visits (N=29 812) |
|---|---|
| | (%*) |
| Caste | |
| Schedule caste (21.4%) | 18.2 |
| Schedule tribe (10.3%) | 13.5 |
| Other backward class (44.0%) | 19.5 |
| None of them/general (24.3%) | 26.4 |
| Urbanicity | |
| Urban (29.6%) | 31.4 |
| Rural (70.4%) | 15.6 |
| Religion | |
| Hindu (79.0%) | 19.8 |
| Muslim (15.9%) | 20.4 |
| Christian (2.1%) | 29.8 |
| Sikh (1.3%) | 22.3 |
| Other (1.6%) | 28.6 |

*The weighted percentage of women with that particular socioeconomic, demographic or media exposure characteristic who attended at least eight antenatal care visits.

least eight ANC visits increased with an increase in the indicators of SES, such as education attainment after complete primary education and wealth. Comparing media exposure characteristics among rural and urban women, each type of media exposure was higher in an urban context. The sample size in each cell fulfilled the conditions to permit the use of multivariable logistic regressions (table 2).

### Media and at least eight ANC visits

After adjusting for age, educational attainment, current marital status, wealth, caste, religion and other media variables, women who had high exposure to newspaper/magazine (aOR, 1.26; 95% CI, 1.18 to 1.35), radio (aOR, 1.29; 95% CI, 1.15 to 1.44) or television (aOR, 1.84; 95% CI, 1.74 to 1.95) or went to cinema to watch a movie at least once a month (aOR, 1.29; 95% CI, 1.19 to 1.4) were more likely to receive at least eight ANC visits compared with those who did not (table 3).

On stratification of populations by urbanicity (table 3), in the urban areas, women who had high exposure to newspaper/magazine (aOR, 1.12; 95% CI, 1.02 to 1.24), radio (aOR, 1.37; 95% CI, 1.13 to 1.65) or television (aOR, 1.39; 95% CI, 1.24 to 1.55) or went to cinema to watch a movie at least once a month (aOR, 1.23; 95% CI, 1.09 to 1.38) were more likely to attend at least eight ANC visits compared with those who did not. In the rural areas, women who had high exposure to newspaper/magazine (aOR, 1.43; 95% CI, 1.31 to

**Table 2** Media exposure characteristics by rural/urban strata among women who gave birth in the last 5 years in India

|  | Urban (N=46 941) | Rural (N=140 953) |
| --- | --- | --- |
|  | %* | %* |
| Total | 100.0 | 100.0 |
| Newspaper/magazine |  |  |
| Low exposure | 78.7 | 94.8 |
| High exposure | 21.3 | 5.3 |
| Radio |  |  |
| Low exposure | 95.1 | 97.3 |
| High exposure | 4.9 | 2.8 |
| Television |  |  |
| Low exposure | 21.0 | 54.9 |
| High exposure | 79.0 | 45.1 |
| Movie |  |  |
| Less than once a month | 85.5 | 95.5 |
| At least once a month | 14.5 | 4.5 |

*The weighted percentage of women in that strata with exposure to the specific type of media with that frequency.

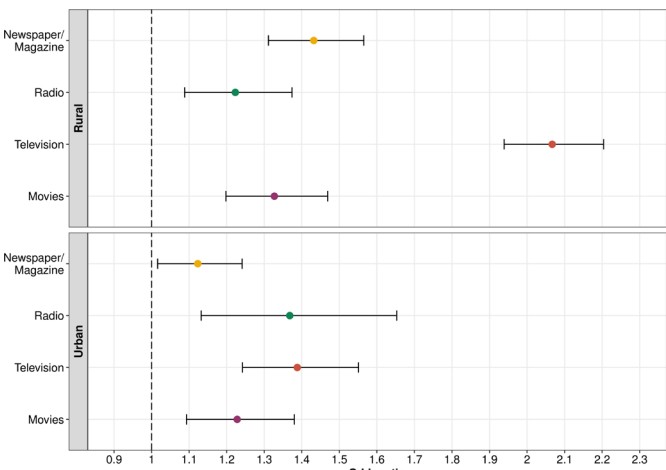

**Figure 1** Association between high exposure to different types of mass media and making at least eight antenatal care visits, stratified by urbanicity (error bars=95% CIs).

1.57), radio (aOR, 1.22; 95% CI, 1.09 to 1.37) or television (aOR, 2.07; 95% CI, 1.94 to 2.2) or went to cinema to watch a movie at least once a month (aOR, 1.33; 95% CI, 1.2 to 1.47) were also more likely to attend at least eight ANC visits compared with those who did not (figure 1).

## DISCUSSION

Media are one of the most valuable communication tools in public health. They have been, time and again, used to create awareness during public health emergencies. Mass media are a powerful tool when the target population is large and spread out, and in-person communication channels are unable to penetrate, especially through underserved rural areas.[25] Various forms of mass media have effectively educated and persuaded audiences to adopt healthy behaviours. Mass media have also been crucial in creating awareness about health-related programmes and campaigns to promote healthy living and improve quality of life.[25]

**Table 3** Crude OR, adjusted aOR (aOR) and aOR stratified by urbanicity, and 95% CIs for the association between media exposure and making at least eight antenatal care visits among women who gave birth in the last 5 years, in the National Family Health Survey 4 (2015–2016)

| At least eight antenatal care visits |  |  | Urban | Rural |
| --- | --- | --- | --- | --- |
|  | Crude OR (95% CI) | aOR* (95% CI) | aOR* (95% CI) | aOR* (95% CI) |
| Newspaper/magazine |  |  |  |  |
| Low exposure (reference) | 1.00 | 1.00 | 1.00 | 1.00 |
| High exposure | 3.14 (2.94 to 3.35) | 1.26 (1.18 to 1.35) | 1.12 (1.02 to 1.24) | 1.43 (1.31 to 1.57) |
| Radio |  |  |  |  |
| Low exposure (reference) | 1.00 | 1.00 | 1.00 | 1.00 |
| High exposure | 2.02 (1.81 to 2.25) | 1.29 (1.15 to 1.44) | 1.37 (1.13 to 1.65) | 1.22 (1.09 to 1.37) |
| Television |  |  |  |  |
| Low exposure (reference) | 1.00 | 1.00 | 1.00 | 1.00 |
| High exposure | 3.92 (3.72 to 4.14) | 1.84 (1.74 to 1.95) | 1.39 (1.24 to 1.55) | 2.07 (1.94 to 2.2) |
| Movie |  |  |  |  |
| Less than once a month (reference) | 1.00 | 1.00 | 1.00 | 1.00 |
| At least once a month | 2.53 (2.34 to 2.74) | 1.29 (1.19 to 1.4) | 1.23 (1.09 to 1.38) | 1.33 (1.2 to 1.47) |

*aORs are mutually adjusted for other media exposure variables as well as age, educational attainment, current marital status, wealth, caste and religion.

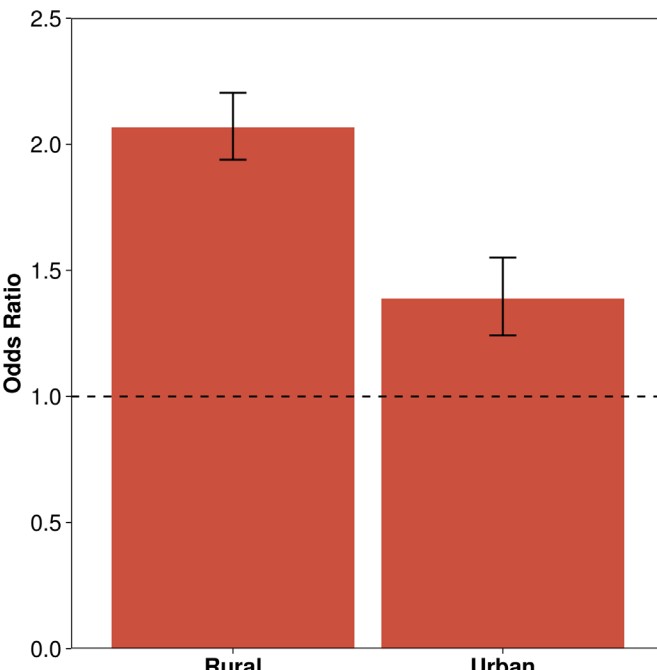

**Figure 2** Association between high exposure to television and making at least eight antenatal care visits, stratified by urbanicity (error bars=95% CIs).

Mass media exposure in South Asian countries is positively associated with utilisation of maternal healthcare services.[26] Through our study, we were able to determine the associations between exposure to different types of mass media and ANC visits among Indian women during pregnancy. The descriptive characteristics established the popularity of different forms of mass media among Indian women. Overall, television appeared to be the most popular media source compared with newspapers/ magazines, radio and watching movies in cinema halls. According to Broadcast Audience Research Council India's recent survey, television is the preferred media of choice in India because of its affordability and its accessibility to free-to-air channels.[27] Furthermore, stratification by urbanicity suggests that high television exposure is more prevalent in urban areas, compared with rural areas. This could be because urban areas in India have better access to facilities like electricity and cable. Other contributing factors could be higher literacy rates in urban areas due to better accessibility of schools and universities.[27 28]

In both rural and urban populations, high exposure to all types of mass media was positively associated with making at least eight ANC visits (figure 1). These findings are in accordance with studies from other developing countries such as Nepal, Bangladesh and Uganda, where they also found positive associations between mass media exposure and ANC visits.[29 30] In Malawi, community-driven mass media campaigns in rural areas resulted in increased uptake of maternal health services including ANC, which suggests that mass media can be an effective mode of intervention in limited-resource settings.[31] Interestingly, we observed that although the exposure to television and other media was higher in urban areas, the associations were stronger in rural areas for all types of media, except radio.

Our study shows that high exposure to newspaper/ magazine is not very prevalent among Indian women, especially in rural areas. However, it is positively associated with making at least eight ANC visits, in both rural and urban regions, after adjusting for several SES indicators. Similarly, a large percentage of both rural and urban women do not have high exposure to radio or go to cinema to watch movies every month. However, the associations of both types of media exposure with making at least eight ANC visits were positive in both rural and urban areas. These findings imply that using newspapers/magazines, radio or movies at cinema halls as communication tools to create awareness about using adequate ANC might have a positive impact, however only on a smaller fraction of women. Alternatively, our data indicate that television is the most popular media source among women, in rural and urban areas. The positive association between television exposure and making at least eight ANC visits suggests that television could be an excellent media source to generate ANC awareness in women, especially in rural areas (figure 2). Previous studies have reported that when television is used as a medium to create knowledge, knowledge gaps resulting from the communication flow of messages are less likely to occur.[32] Our study suggests that using television as a medium to create ANC awareness can have a similar effect on reducing differences in ANC behaviour.

Evidently, television is the most effective media source to create awareness in our large target population. Nonetheless, although only a small percentage of women, in both rural and urban areas, have high exposure to other media sources, it is recommended to use a multimedia approach because, in big countries like India, even a small population percentage is significant.

### Strengths and limitations

A major strength of this study, based on nationally representative survey data, is its generalisability. Furthermore, these data can determine trends stratified by urbanicity, states, SES, caste and religion, which can further help to narrow down the at-risk populations and design customised interventions. Insights from this study can inform health awareness programmes in other developing nations with strong mass media traditions.

One of the main limitations of this study is using self-reported NFHS-4 data, making it prone to recall bias and under-reporting or over-reporting of behaviours. In addition, face-to-face administration of the survey by an interviewer increases the potential for interviewer bias. The cross-sectional design of the data limits the ability to derive any causal inferences on the relationships between the variables. Nonetheless, the strong association between media exposure and ANC visits suggests the potential promise of media in maternal health promotion.

In our study, we determined the associations between media exposure and ANC visits by controlling for

socioeconomic factors such as women's age, education, wealth, current marital status and religion. However, there is a possibility that other factors such as access and resource-related factors, women's autonomy factors (eg, decision-making power and employment status) and partner's socioeconomic factors (eg, education and employment) could also affect the associations between media exposure and making at least eight ANC visits.[26] Future research should examine how the associations between media exposure and ANC visits in India are modified by these additional factors. It is also important to note that, in this study, we only determined the association between the four most common mass media sources and making at least eight ANC visits. However, other modes are also used to create ANC-related awareness in India, such as social media, billboards, cellphones, community theatre, medical camps, door-to-door information by community healthcare workers and pamphlet distributions.[23 33] Moreover, other sources of media such as the internet, computers and smartphones are gaining popularity in India. Future surveys should also include information regarding their use and their implications on health-related behaviours. NFHS-4 has no information about the type of ANC-related media content available to women in India. Thus, one can only speculate based on the literature. Future studies should examine ANC-related media content to obtain a complete picture of the kind of information encouraging such behaviours.

## CONCLUSION

In India, there is a need for increased awareness about obtaining adequate ANC visits. Positive associations between high exposure to different forms of mass media and making at least eight ANC visits provide evidence that exposure to media may encourage positive pregnancy behaviours. In both rural and urban India, the popularity of television and its association with making at least eight ANC visits are stronger than other forms of media. ANC-related health awareness campaigns in the country can use these findings to allocate resources to the appropriate forms of mass media to encourage healthy behaviours. Moreover, clearly, using mass media, especially television, can promote maternal health at a scale, unlike other more labour-intensive approaches.

**Contributors** DD and KV conceived and designed the study. RP and MB contributed to the design of the study. DD analysed and interpreted the data and wrote the first draft of the manuscript. DD, RP, MB and KV contributed to the interpretation of the data and revised the manuscript for important intellectual content. All authors provided critical feedback on the manuscript and agree to be accountable for all aspects of the work.

**Funding** This research was supported by funding from the Bill & Melinda Gates Foundation (Viswanath, PI), grant number INV-005090. The funders had no role in the study design; in the collection, analysis and interpretation of the data; in the writing of the report; and in the decision to submit the paper for publication.

**Competing interests** None declared.

**Patient and public involvement** Patients and/or the public were not involved in the design, or conduct, or reporting, or dissemination plans of this research.

**Patient consent for publication** Not required.

**Ethics approval** The Institutional Review Board of Tufts University School of Medicine/Tufts Medical Center waived the need for ethics approval because this is a cross-sectional study using the publicly available NFHS-4 data set, which did not constitute human subjects research, did not collect identifying information and was intended for quality improvement/quality assurance purposes only.

**Provenance and peer review** Not commissioned; externally peer reviewed.

**Data availability statement** Data are available in a public open access repository. Data are available in public, from the Demographic and Health Surveys Program.

**ORCID iDs**
Dhriti Dhawan http://orcid.org/0000-0003-3005-7772
Ramya Pinnamaneni http://orcid.org/0000-0003-0949-5794
Mesfin Bekalu http://orcid.org/0000-0003-1243-5813
Kasisomayajula Viswanath http://orcid.org/0000-0002-4795-5803

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
