## [Reviewer comments · BMJ Open]

ARTICLE DETAILS

TITLE (PROVISIONAL)	The Association Between Different Types of Mass Media and Antenatal Care Visits in India: A Cross-Sectional Study from The National Family Health Survey (2015-2016)
AUTHORS	Dhawan, Dhriti; Pinnamaneni, Ramya; Bekalu, Mesfin; Viswanath, Kasisomayajula

VERSION 1 – REVIEW

REVIEWER	Rajendra Karkee B.P. Koirala Institute of Health Sciences, Nepal
REVIEW RETURNED	22-Aug-2020

GENERAL COMMENTS	Impression: This article is well written and presented with analysis of the national level survey. The objectives and findings are not impressive anyway because we know that mass media creates awareness; and this has been reported many times and there are many more factors to actually take the delivery services when it comes to save maternal lives. Also, how much feasible is suggesting to have televisions for poor rural families in India? The title should indicate that the data are from the survey. Abstract: Objective: Objective “creating awareness about obtaining adequate Ante Natal Care (ANC) in Indian women” is not matching to the implications/scope of the title. Results: Odds with CI need to be mentioned in the results. Introduction: Background does not explain the ANC utilisation status in India; are not there any publications or national demographic survey to give a figure? Also what is penetration status of mass media and how many households hold mass media like radio, television, and newspaper in rural India? Methods: Exposure variable was categorised as binary: I am concerned how much difference would be there between almost every day and not every day. It is not required to listen the same information every day. I think it makes difference if they are not exposed ‘not at all’. Strength and limitations Are there any sensitive questions to have social bias in this study?
--

REVIEWER	Kaniz Fatema The University of Iowa, Iowa City, Iowa, USA
REVIEW RETURNED	30-Aug-2020

GENERAL COMMENTS	This is really an interesting study in examining the relationship between different types of media exposure and antenatal care (ANC) utilization in Indian women. The objective of this paper is clear in determining how different types of media can play a role in ensuring an adequate number of antenatal visits among pregnant women in India. However, I have a few minor comments regarding the papers. Minor comments:  1. In-Page 6 and line 17, you mentioned “These campaigns frequently associate emotions with achieving change, strengthening the probability of alteration, and increasing the likelihood of adopting new behaviors”. However, there is no mention of the type(s) of the campaign, and what was (were) their message(s)? Please, explain some of them (like how these campaigns succeeded in delivering maternal health messages), then it can make your statement clearer to the readers. 2. Your analysis just focused on ANC. Would you please explain why you didn't include delivery and postnatal care as a measure of childcare? In South-Asian countries, still many women do not deliver their babies by skilled birth attendants even after receiving ANC during their pregnancy (Kaniz & Lariscy 2020). [i] Fatema, K., & Lariscy, J. T. (2020). Mass media exposure and maternal healthcare utilization in South Asia. SSM-Population Health, 100614. 3. Did you control for women's socio-economic status such as women's working status, decision-making power, husband's education, and working status as well? It is widely found in the literature that most women in India are still dominated by their male partners, and they cannot visit the healthcare center without the permission of their husbands (Acharya et al. 2010; Bloom et al. 2001; Biswas, et al. 2017; Mumtaz and Salway, 2007; Umar, 2017). My recommendation is that please include these variables too if your dataset permits. [i] Acharya, D. R., Bell, J. S., Simkhada, P., Van Teijlingen, E. R., & Regmi, P. R. (2010). Women's autonomy in household decision-making: a demographic study in Nepal. Reproductive health, 7(1), 15. [ii] Bloom, S. S., Wypij, D., & Gupta, M. D. (2001). Dimensions of women's autonomy and the influence on maternal health care utilization in a north Indian city. Demography, 38(1), 67-78. [iii] Biswas, A. K., Shovo, T. E. A., Aich, M., & Mondal, S. (2017). Women's autonomy and control to exercise reproductive rights: A sociological study from rural Bangladesh. SAGE Open, 7(2), 2158244017709862. [iv] Mumtaz, Z., & Salway, S. M. (2007). Gender, pregnancy and the uptake of antenatal care services in Pakistan. Sociology of health & illness, 29(1), 1-26. [v] Umar, A. S. (2017). Women Autonomy and the use of Antenatal and Delivery Services in Nigeria. MOJ Public Health, 6(2), 00161. 4. How can ANC reduce the deaths of children under five? Why not postnatal care and immunization? 5. Did you compare the deleted sample characteristics to the analytic sample to see if there was any bias in this approach? 6. The utilization of eight or more antenatal visits is good for both mother's and newborn's baby's health. But, for South-Asian women, receiving eight ANC visits could be burdensome for them
--

	since still many of them live below the poverty line. Would you please explain whether hospitals in India are mostly private or public where women can receive free ANC?
--	--

VERSION 1 – AUTHOR RESPONSE

Reviewer(s)' Comments to Author:

Reviewer: 1

Reviewer Name: Rajendra Karkee

Institution and Country: B.P. Koirala Institute of Health Sciences, Nepal

Please state any competing interests or state 'None declared': None declared

Comments to the Author

Impression:

3. a) This article is well written and presented with analysis of the national level survey. The objectives and findings are not impressive anyway because we know that mass media creates awareness; and this has been reported many times and there are many more factors to actually take the delivery services when it comes to save maternal lives.

b) Also, how much feasible is suggesting to have televisions for poor rural families in India?

Response: We thank the reviewer for his positive as well as constructive comments.

a) Even though other studies have documented that mass media exposure is associated with health-related cognitions and beliefs, our study builds on these in following ways:

- i. While other studies focus on various factors including mass media, in our study, our central focus is on mass media exposure's association with making at least eight antenatal care visits (the new WHO recommendation), while controlling for socio-demographic factors. In literature, there is little or no evidence showing the association between exposure to different types of mass media and health-related behaviors such as utilization of at least eight antenatal care visits, especially in India.
- ii. We believe that while other social determinants are important, mass media exposure is a more readily addressable determinant with appropriate strategic planning as the literature clearly shows.

b) The objective of this paper was to create evidence for the association between different types of mass media and antenatal care visits in India. Suggesting the expansion of television, especially in poor rural families in India is beyond the scope of this study. However, we suggest that, based on the current penetration of different types of mass media in India, creating awareness via television would be most effective in both rural as well as urban India. According to the current statistics based on NFHS-4, overall, 65.2% of Indian households own a television, of which 87% urban households and 53.5% rural households own a television. Moreover, television is a critical mass entertainment medium in both rural and urban India. The Indian television industry has grown in the past few years and is expected to increase in the foreseeable future, in both rural and urban India.[1] It remains the most potent platform for broadest possible reach in urban and rural India.

4. The title should indicate that the data are from the survey.

Response: We thank the reviewer for pointing this out and as suggested, we have revised the title of the study to “The Association Between Different Types of Mass Media and Antenatal Care Visits in India: A Cross-Sectional Study from The National Family Health Survey (2015-2016)”

5. Abstract:

Objective: Objective “creating awareness about obtaining adequate Ante Natal Care (ANC) in Indian women” is not matching to the implications/scope of the title.

Response: We have revised the title and the objective of the study in our manuscript.

6. Results: Odds with CI need to be mentioned in the results.

Response: As suggested by the reviewer and required by the journal's guidelines, we have included the Odds ratios with 95%CI in the result section of the abstract.

7. Introduction: Background does not explain the ANC utilisation status in India; are not there any publications or national demographic survey to give a figure?

Response: As suggested by the reviewer, we have included a figure indicating the ANC utilization status in India to the background section of our study. We have added the following text in the first paragraph under the subheading “Antenatal Care in India”:

“....., only 51.6% pregnant Indian women received four or more ANC visits, which is lower than the global average of 62% during 2010-2016.”

8. Also what is penetration status of mass media and how many households hold mass media like radio, television, and newspaper in rural India?

Response: In rural India, 53.5% rural households own a television and 7% rural households own a radio. Moreover, 66% of women and 80% of men in rural India have regular exposure to some kind of mass media. [2] According to the NFHS-4 report, in rural India, 61.5% women watch television at least once a week, 17.5% women read a newspaper or magazine at least once a week, 8.6% of women listen to the radio at least once a week and 4.7% of the women visit the cinema/theatre at least once a week. On the other hand, in rural India, 70.4% men watch television at least once a week, 46.9% men read a newspaper or magazine at least once a week, 18.1% of men listen to the radio at least once a week, and 16.1% of the men visit the cinema/theatre at least once a week.

To inform the readers about the penetration status and ownership of mass media in India, we have added the following text under the subheading “Mass media Campaigns”:

“In India, 65.2% of households own a television and 8.1% of households own a radio. Moreover, 66% of women in rural India and 92% women in urban India have regular exposure to some kind of mass media.”

9. Methods:

Exposure variable was categorised as binary: I am concerned how much difference would be there between almost every day and not every day. It is not required to listen the same information every day. I think it makes difference if they are not exposed ‘not at all’.

Response:

We agree with the reviewer’s concern as he makes a very valid and legitimate point. In our study, we aimed to determine the association between mass media exposure and making at least eight ANC visits, by comparing high and low exposure to mass media. However, we understand that the

labels “Almost every day” and “Not every day” might be confusing. Literature suggests that behavioral changes through mass media campaigns occur as a result of continuous and cumulative exposure to campaign messages. Such campaigns may require diffusion of messages repeatedly over time to engage the attention of the audience and help them retain information.[3,4] Our assumption is that relatively higher frequency (“Almost every day”) of the use of media compared to lower frequency (“Not every day”), increases the probability of exposure to health information and potentially modify health-related behaviors such as utilization of at least eight antenatal care visits. Accordingly, our findings show that high mass media exposure is strongly associated with higher odds of utilization of at least eight ANC visits. NFHS does not have the data to get into the mechanism which could be explored in other studies.

We have revised the terminology to “High Exposure” (originally labelled “Almost every day”) and “Low Exposure” (originally labelled “Not every day”) in our manuscript.

10. Strength and limitations

Are there any sensitive questions to have social bias in this study?

Response: This particular study did not have sensitive questions per se, but the survey does, especially those related to sexual behaviors and abuse. Therefore, having a social bias refers to the limitations of the survey and not directly to our study.

We have revised our text and removed this limitation from our study.

Reviewer: 2

Reviewer Name: Kaniz Fatema

Institution and Country: The University of Iowa, Iowa City, Iowa, USA

Please state any competing interests or state ‘None declared’: None

Comments to the Author

This is really an interesting study in examining the relationship between different types of media exposure and antenatal care (ANC) utilization in Indian women. The objective of this paper is clear in determining how different types of media can play a role in ensuring an adequate number of antenatal visits among pregnant women in India. However, I have a few minor comments regarding the papers.

Response: We thank the reviewer for her positive and constructive comments.

Minor comments:

11. In-Page 6 and line 17, you mentioned “These campaigns frequently associate emotions with achieving change, strengthening the probability of alteration, and increasing the likelihood of adopting new behaviors”. However, there is no mention of the type(s) of the campaign, and what was (were) their message(s)? Please, explain some of them (like how these campaigns succeeded in delivering maternal health messages), then it can make your statement clearer to the readers.

Response: As per the reviewer’s suggestion, we have added an example of how a maternal health campaign in India succeeded in delivering maternal health messages. This has been added in the same paragraph as mentioned above.

Added text:

" For example, as a part of India's Pradhan Mantri Surakshit Matritva Abhiyan (PMSMA) campaign, public service announcements (PSA) were released. These PSAs have prompted pregnant women as well as their family members to seek ANC during pregnancy, informed them about the benefits of these visits, and highlighted that these visits will be free. The WHO commended India for its groundbreaking progress in reducing maternal mortality and credited the PMSMA campaign for its contribution towards this achievement."

12. Your analysis just focused on ANC. Would you please explain why you didn't include delivery and postnatal care as a measure of childcare? In South-Asian countries, still many women do not deliver their babies by skilled birth attendants even after receiving ANC during their pregnancy (Kaniz & Lariscy 2020).

[i] Fatema, K., & Lariscy, J. T. (2020). Mass media exposure and maternal healthcare utilization in South Asia. *SSM-Population Health*, 100614.

Response:

We agree with the reviewer that factors such as delivery and postnatal care are important measures of childcare. However, in this paper, we focused on ANC because studies have shown that ANC visits play a crucial role in the long-term growth and development of the child.[5] Specifically in South Asia, studies have shown that the frequency of ANC contacts is a determinant of institutional deliveries.[6,7] In India, women who receive at least eight ANC visits are more likely to have an institutional delivery.[7] Therefore, in addition to the factors mentioned by the reviewer, increasing awareness about receiving at least eight ANC visits and the determinants of these visits will add to our knowledge about how to improve birth outcomes.

13. Did you control for women's socio-economic status such as women's working status, decision-making power, husband's education, and working status as well? It is widely found in the literature that most women in India are still dominated by their male partners, and they cannot visit the healthcare center without the permission of their husbands (Acharya et al. 2010; Bloom et al. 2001; Biswas, et al. 2017; Mumtaz and Salway, 2007; Umar, 2017). My recommendation is that please include these variables too if your dataset permits.

[i] Acharya, D. R., Bell, J. S., Simkhada, P., Van Teijlingen, E. R., & Regmi, P. R. (2010). Women's autonomy in household decision-making: a demographic study in Nepal. *Reproductive health*, 7(1), 15.

[ii] Bloom, S. S., Wypij, D., & Gupta, M. D. (2001). Dimensions of women's autonomy and the influence on maternal health care utilization in a north Indian city. *Demography*, 38(1), 67-78.

[iii] Biswas, A. K., Shovo, T. E. A., Aich, M., & Mondal, S. (2017). Women's autonomy and control to exercise reproductive rights: A sociological study from rural Bangladesh. *SAGE Open*, 7(2), 2158244017709862.

[iv] Mumtaz, Z., & Salway, S. M. (2007). Gender, pregnancy and the uptake of antenatal care services in Pakistan. *Sociology of health & illness*, 29(1), 1-26.

[v] Umar, A. S. (2017). Women Autonomy and the use of Antenatal and Delivery Services in Nigeria. *MOJ Public Health*, 6(2), 00161.

Response:

We concur with the reviewer that many factors related to women's socio-demographic status and autonomy (decision-making power) could influence women's utilization of healthcare services in a country like India. In our study, we have controlled for several demographic and socioeconomic factors (specifically, age, education, wealth, current marital status and religion), but we acknowledge that the associations we observed between exposure to mass media and ANC utilization could be

influenced by many other factors related to women's autonomy, as well as access and resource related factors. We have included the below remarks in the "Strength and Limitation" section:

"In our study, we determined the associations between media exposure and ANC visits controlling for socioeconomic factors such as women's age, education, wealth, current marital status, and religion. However, there is a possibility that other factors such as access and resource related factors, women's autonomy factors (for example, decision-making power and employment status), and partner's socio-economic factors (for example, education and employment) could also affect the associations between media exposure and making at least eight ANC visits. Future research should examine how the associations between media exposure and ANC visits in India are modified by these additional factors."

14. How can ANC reduce the deaths of children under five? Why not postnatal care and immunization?

Response:

The association of postnatal care and immunization with mass media can be assessed individually. We are analyzing the same in future papers. In this paper, however we are focusing only on the association of antenatal care visits with mass media. And more specifically, of eight ANC visits, as per the new WHO recommendations.

It has been shown in national surveys and systematic reviews that utilization of ANC services is directly associated with improved birth outcomes and long-term reductions in child mortality and malnourishment.[8] Hence, antenatal visits are a vital investment to improve both maternal and child health outcomes.

15. Did you compare the deleted sample characteristics to the analytic sample to see if there was any bias in this approach?

Response: Yes, we compared the deleted sample characteristics to our analytical sample and there were no biases introduced by taking this approach.

16. The utilization of eight or more antenatal visits is good for both mother's and newborn's baby's health. But, for South-Asian women, receiving eight ANC visits could be burdensome for them since still many of them live below the poverty line. Would you please explain whether hospitals in India are mostly private or public where women can receive free ANC?

Response:

We thank the reviewer for her suggestion, and we have added the following text in the first paragraph under the subheading "Antenatal Care in India", informing the readers about ANC access in India:

"To encourage women to access ANC in India, public healthcare facilities provide these services for free. Moreover, women of low socioeconomic status are given cash incentives for giving birth in a government or an accredited private healthcare facility through schemes such as the Janani Suraksha Yojana (JSY). This scheme also incentivizes healthcare workers to facilitate ANC visits along with institutional births. Studies have shown that JSY has significantly increased ANC visits in India."

Bibliography

- 1 KPMG. India's digital future- Mass of niches. 2019.
<https://assets.kpmg/content/dam/kpmg/in/pdf/2019/08/india-media-entertainment-report-2019.pdf>
 (accessed 28 Oct 2020).
- 2 National Family Health Survey. National Family Health Survey , India. 2017.
<http://rchiips.org/nfhs/NFHS-4Reports/India.pdf> (accessed 10 May 2020).
- 3 Wakefield MA, Loken B, Hornik RC. Use of mass media campaigns to change health behaviour. Lancet 2010;376:1261–71. doi:10.1016/S0140-6736(10)60809-4
- 4 Hornik R, Yanovitzky I. Using theory to design evaluations of communication campaigns: The case of the national youth anti-drug media campaign. Commun Theory 2003;13:204–24.
 doi:10.1111/j.1468-2885.2003.tb00289.x
- 5 Kumar G, Choudhary TS, Srivastava A, et al. Utilisation, equity and determinants of full antenatal care in India: analysis from the National Family Health Survey 4. BMC Pregnancy Childbirth 2019;19:327. doi:10.1186/s12884-019-2473-6
- 6 Fatema K, Lariscy JT. Mass media exposure and maternal healthcare utilization in South Asia. SSM - Popul Heal 2020;11. doi:10.1016/j.ssmph.2020.100614
- 7 Paul PL, Pandey S. Factors influencing institutional delivery and the role of accredited social health activist (ASHA): a secondary analysis of India human development survey 2012. BMC Pregnancy Childbirth 2020;20:445. doi:10.1186/s12884-020-03127-z
- 8 Kuhnt J, Vollmer S. Antenatal care services and its implications for vital and health outcomes of children: Evidence from 193 surveys in 69 low-income and middle-income countries. BMJ Open 2017;7. doi:10.1136/bmjopen-2017-017122

VERSION 2 – REVIEW

REVIEWER	Rajendra Karkee B.P. Koirala Institute of Health Sciences Nepal
REVIEW RETURNED	26-Nov-2020

GENERAL COMMENTS	I thank authors for their revision and response.
--

REVIEWER	Kaniz Fatema The University of Iowa, Iowa City, USA
REVIEW RETURNED	28-Nov-2020

GENERAL COMMENTS	I am satisfied with the updated version.
--